# Mapping the global geography of cybercrime with the World Cybercrime Index

**Miranda Bruce** [1,2]*, **Jonathan Lusthaus**[1,3], **Ridhi Kashyap**[1,4], **Nigel Phair**[5], **Federico Varese**[6]

1 Department of Sociology, University of Oxford, Oxford, United Kingdom, 2 Canberra School of Professional Studies, University of New South Wales, Canberra, Australia, 3 Oxford School of Global and Area Studies, University of Oxford, Oxford, United Kingdom, 4 Leverhulme Centre for Demographic Science, University of Oxford, Oxford, United Kingdom, 5 Department of Software Systems and Cybersecurity, Faculty of IT, Monash University, Victoria, Australia, 6 Centre d'études européennes et de politique comparée, Sciences Po, Paris, France

* miranda.bruce@sociology.ox.ac.uk

## Abstract

Cybercrime is a major challenge facing the world, with estimated costs ranging from the hundreds of millions to the trillions. Despite the threat it poses, cybercrime is somewhat an invisible phenomenon. In carrying out their virtual attacks, offenders often mask their physical locations by hiding behind online nicknames and technical protections. This means technical data are not well suited to establishing the true location of offenders and scholarly knowledge of cybercrime geography is limited. This paper proposes a solution: an expert survey. From March to October 2021 we invited leading experts in cybercrime intelligence/investigations from across the world to participate in an anonymized online survey on the geographical location of cybercrime offenders. The survey asked participants to consider five major categories of cybercrime, nominate the countries that they consider to be the most significant sources of each of these types of cybercrimes, and then rank each nominated country according to the impact, professionalism, and technical skill of its offenders. The outcome of the survey is the World Cybercrime Index, a global metric of cybercriminality organised around five types of cybercrime. The results indicate that a relatively small number of countries house the greatest cybercriminal threats. These findings partially remove the veil of anonymity around cybercriminal offenders, may aid law enforcement and policymakers in fighting this threat, and contribute to the study of cybercrime as a local phenomenon.

## Introduction

Although the geography of cybercrime attacks has been documented, the geography of cybercrime offenders–and the corresponding level of "cybercriminality" present within each country–is largely unknown. A number of scholars have noted that valid and reliable data on offender geography are sparse [1–4], and there are several significant obstacles to establishing a robust metric of cybercriminality by country. First, there are the general challenges associated with the study of any hidden population, for whom no sampling frame exists [5, 6]. If cybercriminals themselves cannot be easily accessed or reliably surveyed, then cybercriminality

**Data Availability Statement:** The dataset and relevant documents have been uploaded to the Open Science Framework. Data can be accessed via the following URL: https://osf.io/5s72x/?view_only=ea7ee238f3084054a6433fbab43dc9fb.

**Funding:** This project has received funding from the European Research Council (ERC) under the European Union's Horizon 2020 research and innovation program (Grant agreement No. 101020598 – CRIMGOV, Federico Varese PI). FV received the award and is the Primary Investigator. The ERC did not play any role in the study design, data collection and analysis, decision to publish, or preparation of the manuscript. Funder website: https://erc.europa.eu/faq-programme/h2020.

**Competing interests:** The authors have declared that no competing interests exist.

must be measured through a proxy. This is the second major obstacle: deciding what kind of proxy data would produce the most valid measure of cybercriminality. While there is much technical data on cybercrime attacks, this data captures artefacts of the digital infrastructure or proxy (obfuscation) services used by cybercriminals, rather than their true physical location. Non-technical data, such as legal cases, can provide geographical attribution for a small number of cases, but the data are not representative of global cybercrime. In short, the question of how best to measure the geography of cybercriminal offenders is complex and unresolved.

There is tremendous value in developing a metric for cybercrime. Cybercrime is a major challenge facing the world, with the most sober cost estimates in the hundreds of millions [7, 8], but with high-end estimates in the trillions [9]. By accurately identifying which countries are cybercrime hotspots, the public and private sectors could concentrate their resources on these hotspots and spend less time and funds on cybercrime countermeasures in countries where the problem is limited. Whichever strategies are deployed in the fight against cybercrime (see for example [10–12]), they should be targeted at countries that produce the largest cybercriminal threat [3]. A measure of cybercriminality would also enable other lines of scholarly inquiry. For instance, an index of cybercriminality by country would allow for a genuine dependent variable to be deployed in studies attempting to assess which national characteristics–such as educational attainment, Internet penetration, or GDP–are associated with cybercrime [4, 13]. These associations could also be used to identify future cybercrime hubs so that early interventions could be made in at-risk countries before a serious cybercrime problem develops. Finally, this metric would speak directly to theoretical debates on the locality of cybercrime, and organized crime more generally [11–14]. The challenge we have accepted is to develop a metric that is both global and robust. The following sections respectively outline the background elements of this study, the methods, the results, and then discussion and limitations.

## Background

Profit-driven cybercrime, which is the focus of this paper/research, has been studied by both social scientists and computer scientists. It has been characterised by empirical contributions that have sought to illuminate the nature and organisation of cybercrime both online and offline [15–20]. But, as noted above, the geography of cybercrime has only been addressed by a handful of scholars, and they have identified a number of challenges connected to existing data. In a review of existing work in this area, Lusthaus et al. [2] identify two flaws in existing cybercrime metrics: 1) their ability to correctly attribute the location of cybercrime offenders; 2) beyond a handful of examples, their ability to compare the severity and scale of cybercrime between countries.

Building attribution into a cybercrime index is challenging. Often using technical data, cybersecurity firms, law enforcement agencies and international organisations regularly publish reports that identify the major sources of cyber attacks (see for example [21–24]). Some of these sources have been aggregated by scholars (see [20, 25–29]). But the kind of technical data contained in these reports cannot accurately measure offender location. Kigerl [1] provides some illustrative remarks:

> Where the cybercriminals live is not necessarily where the cyberattacks are coming from. An offender from Romania can control zombies in a botnet, mostly located in the United States, from which to send spam to countries all over the world, with links contained in them to phishing sites located in China. The cybercriminal's reach is not limited by national borders
>
> (p. 473).

As cybercriminals often employ proxy services to hide their IP addresses, carry out attacks across national boundaries, collaborate with partners around the world, and can draw on infrastructure based in different countries, superficial measures do not capture the true geographical distribution of these offenders. Lusthaus et al. [2] conclude that attempts to produce an index of cybercrime by country using technical data suffer from a problem of validity. "If they are a measure of anything", they argue, "they are a measure of cyber-attack geography", not of the geography of offenders themselves (p. 452).

Non-technical data are far better suited to incorporating attribution. Court records, indictments and other investigatory materials speak more directly to the identification of offenders and provide more granular detail on their location. But while this type of data is well matched to micro-level analysis and case studies, there are fundamental questions about the representativeness of these small samples, even if collated. First, any sample would capture cases only where cybercriminals had been prosecuted, and would not include offenders that remain at large. Second, if the aim was to count the number of cybercrime prosecutions by country, this may reflect the seriousness with which various countries take cybercrime law enforcement or the resources they have to pursue it, rather than the actual level of cybercrime within each country (for a discussion see [30, 31]). Given such concerns, legal data is also not an appropriate approach for such a research program.

Furthermore, to carry out serious study on this topic, a cybercrime metric should aim to include as many countries as possible, and the sample must allow for variation so that high and low cybercrime countries can be compared. If only a handful of widely known cybercrime hubs are studied, this will result in selection on the dependent variable. The obvious challenge in providing such a comparative scale is the lack of good quality data to devise it. As an illustration, in their literature review Hall et al. [10] identify the "dearth of robust data" on the geographical location of cybercriminals, which means they are only able to include six countries in their final analysis (p. 285. See also [4, 32, 33]).

Considering the weaknesses within both existing technical and legal data discussed above, Lusthaus et al. [2] argue for the use of an expert survey to establish a global metric of cybercriminality. Expert survey data "can be extrapolated and operationalised", and "attribution can remain a key part of the survey, as long as the participants in the sample have an extensive knowledge of cybercriminals and their operations" (p. 453). Up to this point, no such study has been produced. Such a survey would need to be very carefully designed for the resulting data to be both reliable and valid. One criticism of past cybercrime research is that surveys were used whenever other data was not immediately available, and that they were not always designed with care (for a discussion see [34]).

## Methods

In response to the preceding considerations, we designed an expert survey in 2020, refined it through focus groups, and deployed it throughout 2021. The survey asked participants to consider five major types of cybercrime–*Technical products/services*; *Attacks and extortion*; *Data/ identity theft*; *Scams*; and *Cashing out/money laundering*–and nominate the countries that they consider to be the most significant sources of each of these cybercrime types. Participants then rated each nominated country according to the impact of the offenses produced there, and the professionalism and technical skill of the offenders based there. Using the expert responses, we generated scores for each type of cybercrime, which we then combined into an overall metric of cybercriminality by country: the World Cybercrime Index (WCI). The WCI achieves our initial goal to devise a valid measure of cybercrime hub location and significance, and is the

first step in our broader aim to understand the local dimensions of cybercrime production across the world.

## Participants

Identifying and recruiting cybercrime experts is challenging. Much like the hidden population of cybercriminals we were trying to study, cybercrime experts themselves are also something of a hidden population. Due to the nature of their work, professionals working in the field of cybercrime tend to be particularly wary of unsolicited communication. There is also the problem of determining who is a true cybercrime expert, and who is simply presenting themselves as one. We designed a multi-layered sampling method to address such challenges.

The heart of our strategy involved purposive sampling. For an index based entirely on expert opinion, ensuring the quality of these experts (and thereby the quality of our survey results) was of the utmost importance. We defined "expertise" as adult professionals who have been engaged in cybercrime intelligence, investigation, and/or attribution for a minimum of five years and had a reputation for excellence amongst their peers. Only currently- or recently-practicing intelligence officers and investigators were included in the participant pool. While participants could be from either the public or private sectors, we explicitly excluded professionals working in the field of cybercrime research who are not actively involved in tracking offenders, which includes writers and academics. In short, only experts with first-hand knowledge of cybercriminals are included in our sample. To ensure we had the leading experts from a wide range of backgrounds and geographical areas, we adopted two approaches for recruitment. We searched extensively through a range of online sources including social media (e.g. LinkedIn), corporate sites, news articles and cybercrime conference programs to identify individuals who met our inclusion criteria. We then faced a second challenge of having to find or discern contact information for these individuals.

Complementing this strategy, the authors also used their existing relationships with recognised cybercrime experts to recruit participants using the "snowball" method [35]. This both enhanced access and provided a mechanism for those we knew were bona fide experts to recommend other bona fide experts. The majority of our participants were recruited in this manner, either directly through our initial contacts or through a series of referrals that followed. But it is important to note that this snowball sampling fell under our broader purposive sampling strategy. That is, all the original "seeds" had to meet our inclusion criteria of being a top expert in the first instance. Any connections we were offered also had to meet our criteria or we would not invite them to participate. Another important aspect of this sampling strategy is that we did not rely on only one gatekeeper, but numerous, often unrelated, individuals who helped us with introductions. This approach reduced bias in the sample. It was particularly important to deploy a number of different "snowballs" to ensure that we included experts from each region of the world (Africa, Asia Pacific, Europe, North America and South America) and from a range of relevant professional backgrounds. We limited our sampling strategy to English speakers. The survey itself was likewise written in English. The use of English was partly driven by the resources available for this study, but the population of cybercrime experts is itself very global, with many attending international conferences and cooperating with colleagues from across the world. English is widely spoken within this community. While we expect the gains to be limited, future surveys will be translated into some additional languages (e.g. Spanish and Chinese) to accommodate any non-English speaking experts that we may not otherwise be able to reach.

Our survey design, detailed below, received ethics approval from the Human Research Advisory Panel (HREAP A) at the University of New South Wales in Australia, approval

number HC200488, and the Research Ethics Committee of the Department of Sociology (DREC) at the University of Oxford in the United Kingdom, approval number SOC_R2_001_C1A_20_23. Participants were recruited in waves between 1 August 2020 and 30 September 2021. All participants provided consent to participate in the focus groups, pilot survey, and final survey.

## Survey design

The survey comprised three stages. First, we conducted three focus groups with seven experts in cybercrime intelligence/investigations to evaluate our initial assumptions, concepts, and framework. These experts were recruited because they had reputations as some of the very top experts in the field; they represented a range of backgrounds in terms of their own geographical locations and expertise across different types of cybercrime; and they spanned both the public and private sectors. In short, they offered a cross-section of the survey sample we aimed to recruit. These focus groups informed several refinements to the survey design and specific terms to make them better comprehensible to participants. Some of the key terms, such as "professionalism" and "impact", were a direct result of this process. Second, some participants from the focus groups then completed a pilot version of the survey, alongside others who had not taken part in these focus groups, who could offer a fresh perspective. This allowed us to test technical components, survey questions, and user experience. The pilot participants provided useful feedback and prompted a further refinement of our approach. The final survey was released online in March 2021 and closed in October 2021. We implemented several elements to ensure data quality, including a series of preceding statements about time expectations, attention checks, and visual cues throughout the survey. These elements significantly increased the likelihood that our participants were both suitable and would provide full and thoughtful responses.

The introduction to the survey outlined the survey's two main purposes: to identify which countries are the most significant sources of profit-driven cybercrime, and to determine how impactful the cybercrime is in these locations. Participants were reminded that state-based actors and offenders driven primarily by personal interests (for instance, cyberbullying or harassment) should be excluded from their consideration. We defined the "source" of cybercrime as the country where offenders are primarily based, rather than their nationality. To maintain a level of consistency, we made the decision to only include countries formally recognised by the United Nations. We initially developed seven categories of cybercrime to be included in the survey, based on existing research. But during the focus groups and pilot survey, our experts converged on five categories as the most significant cybercrime threats on a global scale:

1. Technical products/services (e.g. malware coding, botnet access, access to compromised systems, tool production).

2. Attacks and extortion (e.g. DDoS attacks, ransomware).

3. Data/identity theft (e.g. hacking, phishing, account compromises, credit card comprises).

4. Scams (e.g. advance fee fraud, business email compromise, online auction fraud).

5. Cashing out/money laundering (e.g. credit card fraud, money mules, illicit virtual currency platforms).

After being prompted with these descriptions and a series of images of world maps to ensure participants considered a wide range of regions/countries, participants were asked to

nominate up to five countries that they believed were the most significant sources of each of these types of cybercrime. Countries could be listed in any order; participants were not instructed to rank them. Nominating countries was optional and participants were free to skip entire categories if they wished. Participants were then asked to rate each of the countries they nominated against three measures: how impactful the cybercrime is, how professional the cybercrime offenders are, and how technically skilled the cybercrime offenders are. Across each of these three measures, participants were asked to assign scores on a Likert-type scale between 1 (e.g. least professional) to 10 (e.g. most professional). Nominating and then rating countries was repeated for all five cybercrime categories.

This process, of nominating and then rating countries across each category, introduces a potential limitation in the survey design: the possibility of survey response fatigue. If a participant nominated the maximum number of countries across each cybercrime category– 25 countries–by the end of the survey they would have completed 75 Likert-type scales. The repetition of this task, paired with the consideration that it requires, has the potential to introduce respondent fatigue as the survey progresses, in the form of response attrition, an increase in careless responses, and/or increased likelihood of significantly higher/lower scores given. This is a common phenomenon in long-form surveys [36], and especially online surveys [37, 38]. Jeong et al [39], for instance, found that questions asked near the end of a 2.5 hour survey were 10–64% more likely to be skipped than those at the beginning. We designed the survey carefully, refined with the aid of focus groups and a pilot, to ensure that only the most essential questions were asked. As such, the survey was not overly long (estimated to take 30 minutes). To accommodate any cognitive load, participants were allowed to complete the survey anytime within a two-week window. Their progress was saved after each session, which enabled participants to take breaks between completing each section (a suggestion made by Jeong et al [39]). Crucially, throughout survey recruitment, participants were informed that the survey is time-intensive and required significant attention. At the beginning of the survey, participants were instructed not to undertake the survey unless they could allocate 30 minutes to it. This approach pre-empted survey fatigue by discouraging those likely to lose interest from participating. This compounds the fact that only experts with a specific/strong interest in the subject matter of the survey were invited to participate. Survey fatigue is addressed further in the Discussion section, where we provide an analysis suggesting little evidence of participant fatigue.

In sum, we designed the survey to protect against various sources of bias and error, and there are encouraging signs that the effects of these issues in the data are limited (see Discussion). Yet expert surveys are inherently prone to some types of bias and response issues; in the WCI, the issue of selection and self-selection within our pool of experts, as well as geo-political biases that may lead to systematic over- or under-scoring of certain countries, is something we considered closely. We discuss these issues in detail in the subsection on Limitations below.

**Measures.** Using the survey responses, we define the following two metrics: (i) a cybercriminality "type" score for each of the five crime types; (ii) an "overall" score across all types of cybercrime, which we term the World Cybercrime Index (WCI). We calculate the cybercriminality score for each crime type–the $WCI_{type}$ score–in two steps. First, we first calculate the average score across the three dimensions (impact, professionalism and technical skill) across all nominations for that country within one of the five cybercrime types. The average score of each measure is then averaged into a "type" score for each country, as shown in Eq (1):

$$Country\ Score_{type} = \frac{1}{nominations} \sum_{i=1}^{nominations} \frac{(I + P + TS)}{3} \tag{1}$$

This "type" score is then multiplied by the proportion of experts who nominated that country. Within each cybercrime type, a country could be nominated a possible total of 92 times–once per participant. We then multiply this weighted score by ten to produce a continuous scale out of 100 (see Eq (2)). This process prevents countries that received high scores, but a low number of nominations, from receiving artificially high rankings.

$$WCI_{type} = Country\ Score_{type} * \left(\frac{nominations}{92}\right) * 10 \tag{2}$$

We calculate the WCI$_{overall}$ score for each country using a similar process. First, we calculate the country's average score (Country Score$_{type}$ from Eq 1) for all five cybercrime types. We then average these five type scores together into an overall score. This overall score is then multiplied by the sum of nominations across all crime types, divided by the total possible nominations for each country, which is increased to 460 (once per 92 participants, per 5 cybercrime types). This score is then multiplied by ten to produce a continuous scale out of 100, as shown in Eq (3):

$$WCI_{overall} = \left(\frac{1}{5}\right) * \sum_{i=1}^{type} * \left(\frac{\sum_{i=1}^{type} nominations}{460}\right) * 10 \tag{3}$$

The analyses for this paper were performed in R. All data and code have been made publicly available so that our analysis can be reproduced and extended.

## Results

We contacted 245 individuals to participate in the survey, of which 147 agreed and were sent invitation links to participate. Out of these 147, a total of 92 people completed the survey, giving us an overall response rate of 37.5%. Given the expert nature of the sample, this is a high response rate (for a detailed discussion see [40]), and one just below what Wu, Zhao, and Fils-Aime estimate of response rates for general online surveys in social science: 44% [41]. The survey collected information on the participants' primary nationality and their current country of residence. Four participants chose not to identify their nationality. Overall, participants represented all five major geopolitical regions (Africa, the Asia-Pacific, Europe, North America and South America), both in nationality and residence, though the distribution was uneven and concentrated in particular regions/countries. There were 8 participants from Africa, 11 participants from the Asia Pacific, 27 from North America, and 39 from Europe. South America was the least represented region with only 3 participants. A full breakdown of participants' nationality, residence, and areas of expertise is included in the Supporting Information document (see S1 Appendix).

Table 1 shows the scores for the top fifteen countries of the WCI$_{overall}$ index. Each entry shows the country, along with the mean score (out of 10) averaged across the participants who nominated this country, for three categories: impact, professionalism, and technical skill. This is followed by each country's WCI$_{overall}$ and WCI$_{type}$ scores. Countries are ordered by their WCI$_{overall}$ score. Each country's highest WCI$_{type}$ scores are highlighted. Full indices that include all 197 UN-recognised countries can be found in S1 Indices.

Some initial patterns can be observed from this table, as well as the full indices in the supplementary document (see S1 Indices). First, a small number of countries hold consistently high ranks for cybercrime. Six countries–China, Russia, Ukraine, the US, Romania, and Nigeria–appear in the top 10 of every WCI$_{type}$ index, including the WCI$_{overall}$ index. Aside from

**Table 1. World Cybercrime Index overall–top 15 countries.**

| Rank | Country | I | P | TS | WCI Score | Tech | Attacks | Data | Scams | Cash |
|------|---------|-----|-----|-----|-----------|-------|---------|-------|-------|-------|
| 1 | Russia | 8.96 | 8.81 | 8.73 | **58.39** | 82.17 | 81.34 | 65.18 | 21.70 | 41.56 |
| 2 | Ukraine | 8.37 | 8.29 | 8.24 | **36.44** | 52.97 | 50.76 | 36.01 | 11.20 | 31.27 |
| 3 | China | 8.22 | 7.70 | 7.81 | **27.86** | 40.22 | 24.24 | 34.89 | 15.83 | 24.13 |
| 4 | United States | 7.99 | 7.21 | 7.21 | **25.01** | 27.64 | 17.68 | 30.36 | 22.72 | 26.63 |
| 5 | Nigeria | 8.25 | 6.49 | 5.80 | **21.28** | 7.93 | 8.41 | 23.04 | 52.17 | 14.86 |
| 6 | Romania | 7.12 | 7.04 | 7.15 | **14.83** | 17.83 | 9.17 | 22.50 | 13.15 | 11.49 |
| 7 | North Korea | 7.91 | 7.23 | 7.38 | **10.61** | 8.66 | 25.33 | 13.01 | 2.17 | 3.88 |
| 8 | United Kingdom | 7.86 | 7.21 | 6.75 | **9.01** | 5.04 | 4.75 | 5.80 | 7.86 | 21.63 |
| 9 | Brazil | 6.90 | 6.35 | 6.32 | **8.93** | 13.70 | 8.77 | 10.29 | 7.28 | 4.64 |
| 10 | India | 7.90 | 6.60 | 6.65 | **6.13** | 4.46 | 3.62 | 6.81 | 12.75 | 3.01 |
| 11 | Iran | 6.88 | 6.45 | 6.64 | **4.78** | 8.62 | 10.00 | 3.59 | 0.94 | 0.72 |
| 12 | Belarus | 6.84 | 7.20 | 7.32 | **3.87** | 11.92 | 5.58 | 1.85 | - - | - - |
| 13 | Ghana | 8.57 | 6.83 | 6.09 | **3.58** | 1.23 | 0.76 | 2.97 | 10.36 | 2.57 |
| 14 | South Africa | 6.95 | 5.35 | 5.50 | **2.58** | 1.20 | 0.65 | 0.58 | 7.17 | 3.30 |
| 15 | Moldova | 7.38 | 7.19 | 7.56 | **2.57** | 6.70 | 0.98 | 2.43 | 0.83 | 1.88 |

I = Impact; P = Professionalism; TS = Technical skill, Technical = *Technical products/services*, Attacks = *Attacks and extortion*, Data = *Data/identity theft*, Cash = *Cashing out and money laundering*. I, P, and TS are scored out of 10. 'WCI Score', and all columns following, are scored out of 100. Each country's top score across all cybercrime types is shaded in grey.

Romania, all appear in the top three at least once. While appearing in a different order, the first ten countries in the *Technical products/services* and *Attacks and extortion* indices are the same. Second, despite this small list of countries regularly appearing as cybercrime hubs, the survey results capture a broad geographical diversity. All five geopolitical regions are represented across each type. Overall, 97 distinct countries were nominated by at least one expert. This can be broken down into the cybercrime categories. *Technical products/services* includes 41 different countries; *Attacks and extortion* 43; *Data/identity theft* 51; *Scams* 49; and *Cashing out/money laundering* 63.

Some key findings emerge from these results, which are further illustrated by the following Figs 1 and 2. First, cybercrime is not universally distributed. Certain countries are cybercrime hubs, while many others are not associated with cybercriminality in a serious way. Second, countries that are cybercrime hubs specialise in particular types of cybercrime. That is, despite a small number of countries being leading producers of cybercrime, there is meaningful variation between them both across categories, and in relation to scores for impact, professionalism and technical skill. Third, the results show a longer list of cybercrime-producing countries than are usually included in publications on the geography of cybercrime. As the survey captures leading producers of cybercrime, rather than just any country where cybercrime is present, this suggests that, even if a small number of countries are of serious concern, and close to 100 are of little concern at all, the remaining half are of at least moderate concern.

To examine further the second finding concerning hub specialisation, we calculated an overall "Technicality score"–or "T-score"–for the top 15 countries of the $WCI_{overall}$ index. We assigned a value from 2 to -2 to each type of cybercrime to designate the level of technical complexity involved. *Technical products/services* is the most technically complex type (2), followed by *Attacks and extortion* (1), *Data/identity theft* (0), *Scams* (-1), and finally *Cashing out and money laundering* (-2), which has very low technical complexity. We then multiplied each country's WCI score for each cybercrime type by its assigned value–for instance, a *Scams* WCI

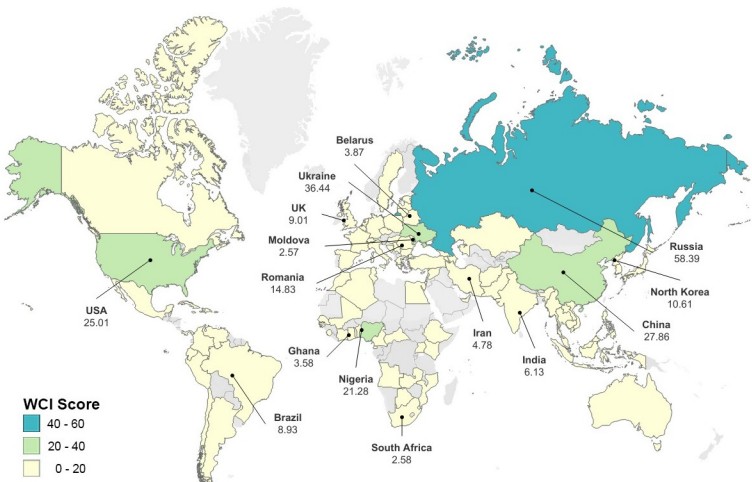

**Fig 1. World map of the WCI_overall index–top 15 countries labelled.** Base map and data from OpenStreetMap and OpenStreetMap Foundation.

score of 5 would be multiplied by -1, with a final modified score of -5. As a final step, for each country, we added all of their modified WCI scores across all five categories together to generate the T-score. Fig 3 plots the top 15 WCI_overall countries' T-scores, ordering them by score. Countries with negative T-scores are highlighted in red, and countries with positive scores are in black.

The T-score is best suited to characterising a given hub's specialisation. For instance, as the line graph makes clear, Russia and Ukraine are highly technical cybercrime hubs, whereas Nigerian cybercriminals are engaged in less technical forms of cybercrime. But for countries that lie close to the centre (0), the story is more complex. Some may specialise in cybercrime

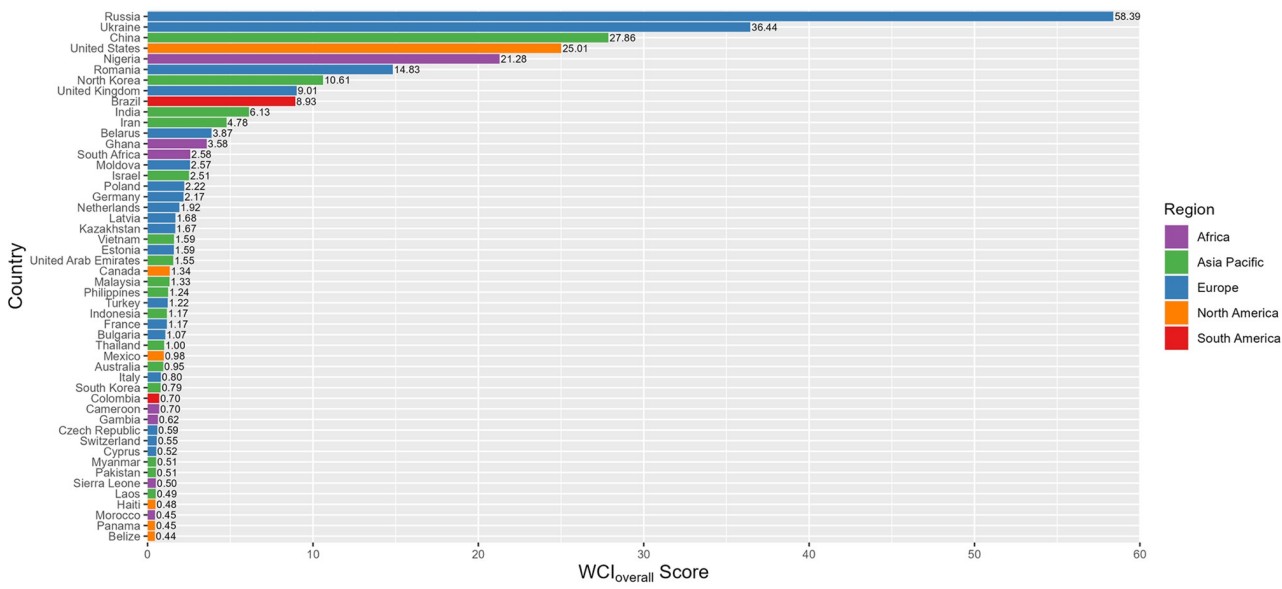

**Fig 2. Top 50 countries by WCI_overall score.**

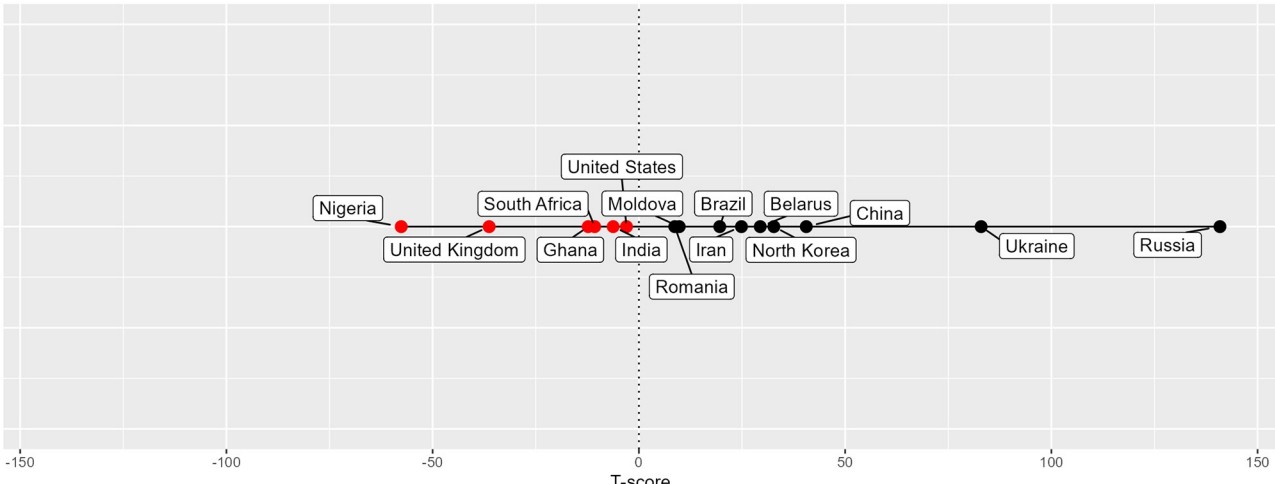

**Fig 3. Technicality or T-score for the top 15 WCI$_{overall}$ countries.** Negative values correspond to lower technicality, positive values to higher technicality.

types with middling technical complexity (e.g. *Data/identity theft*). Others may specialise in both high- and low-tech crimes. In this sample of countries, India (-6.02) somewhat specialises in *Scams* but is otherwise a balanced hub, whereas Romania (10.41) and the USA (-2.62) specialise in both technical and non-technical crimes, balancing their scores towards zero. In short, each country has a distinct profile, indicating a unique local dimension.

## Discussion

This paper introduces a global and robust metric of cybercriminality–the World Cybercrime Index. The WCI moves past previous technical measures of cyber attack geography to establish a more focused measure of the geography of cybercrime offenders. Elicited through an expert survey, the WCI shows that cybercrime is not universally distributed. The key theoretical contribution of this index is to illustrate that cybercrime, often seen as a fluid and global type of organized crime, actually has a strong local dimension (in keeping with broader arguments by some scholars, such as [14, 42]).

While we took a number of steps to ensure our sample of experts was geographically representative, the sample is skewed towards some regions (such as Europe) and some countries (such as the US). This may simply reflect the high concentration of leading cybercrime experts in these locations. But it is also possible this distribution reflects other factors, including the authors' own social networks; the concentration of cybercrime taskforces and organisations in particular countries; the visibility of different nations on networking platforms like LinkedIn; and also perhaps norms of enthusiasm or suspicion towards foreign research projects, both inside particular organisations and between nations.

To better understand what biases might have influenced the survey data, we analysed participant rating behaviours with a series of linear regressions. Numerical ratings were the response and different participant characteristics–country of nationality; country of residence; crime type expertise; and regional expertise–were the predictors. Our analysis found evidence ($p < 0.05$) that participants assigned higher ratings to the countr(ies) they either reside in or are citizens of, though this was not a strong or consistent result. For instance, regional experts did not consistently rate their region of expertise more highly than other regions. European

and North American experts, for example, rated countries from these regions lower than countries from other regions. Our analysis of cybercrime type expertise showed even less systematic rating behaviour, with no regression yielding a statistically significant ($p < 0.05$) result. Small sample sizes across other known participant characteristics meant that further analyses of rating behaviour could not be performed. This applied to, for instance, whether residents and citizens of the top ten countries in the WCI nominated their own countries more or less often than other experts. On this point: 46% of participants nominated their own country at some point in the survey, but the majority (83%) of nominations were for a country different to the participant's own country of residence or nationality. This suggested limited bias towards nominating one's own country. Overall, these analyses point to an encouraging observation: while there is a slight home-country bias, this does not systematically result in higher rating behaviour. Longitudinal data from future surveys, as well as a larger participant pool, will better clarify what other biases may affect rating behaviour.

There is little evidence to suggest that survey fatigue affected our data. As the survey progressed, the heterogeneity of nominated countries across all experts increased, from 41 different countries nominated in the first category to 63 different countries nominated in the final category. If fatigue played a significant role in the results then we would expect this number to decrease, as participants were not required to nominate countries within a category and would have been motivated to nominate fewer countries to avoid extending their survey time. We further investigated the data for evidence of survey fatigue in two additional ways: by performing a Mann-Kendall/Sen's slope trend test (MK/S) to determine whether scores skewed significantly upwards or downwards towards the end of the survey; and by compiling an intra-individual response variability (IRV) index to search for long strings of repeated scores at the end of the survey [43]. The MK/S test was marginally statistically significant ($p < 0.048$), but the results indicated that scores trended downwards only minimally (-0.002 slope coefficient). Likewise, while the IRV index uncovered a small group of participants (n = 5) who repeatedly inserted the same score, this behaviour was not more likely to happen at the end of the survey (see S7 and S8 Tables in S1 Appendix).

It is encouraging that there is at least some external validation for the WCI's highest ranked countries. Steenbergen and Marks [44] recommend that data produced from expert judgements should "demonstrate convergent validity with other measures of [the topic]–that is, the experts should provide evaluations of the same [. . .] phenomenon that other measurement instruments pick up." (p. 359) Most studies of the global cybercrime geography are, as noted in the introduction, based on technical measures that cannot accurately establish the true physical location of offenders (for example [1, 4, 28, 33, 45]). Comparing our results to these studies would therefore be of little value, as the phenomena being measured differs: they are measuring attack infrastructure, whereas the WCI measures offender location. Instead, looking at in-depth qualitative cybercrime case studies would provide a better comparison, at least for the small number of higher ranked countries. Though few such studies into profit-driven cybercrime exist, and the number of countries included are limited, we can see that the top ranked countries in the WCI match the key cybercrime producing countries discussed in the qualitative literature (see for example [3, 10, 32, 46–50]). Beyond this qualitative support, our sampling strategy–discussed in the Methods section above–is our most robust control for ensuring the validity of our data.

Along with contributing to theoretical debates on the (local) nature of organized crime [1, 14], this index can also contribute to policy discussions. For instance, there is an ongoing debate as to the best approaches to take in cybercrime reduction, whether this involves improving cyber-law enforcement capacity [3, 51], increasing legitimate job opportunities and access to youth programs for potential offenders [52, 53], strengthening international

agreements and law harmonization [54–56], developing more sophisticated and culturally-specific social engineering countermeasures [57], or reducing corruption [3, 58]. As demonstrated by the geographical, economic, and political diversity of the top 15 countries (see Table 1), the likelihood that a single strategy will work in all cases is low. If cybercrime is driven by local factors, then mitigating it may require a localised approach that considers the different features of cybercrime in these contexts. But no matter what strategies are applied in the fight against cybercrime, they should be targeted at the countries that produce the most cybercrime, or at least produce the most impactful forms of it [3]. An index is a valuable resource for determining these countries and directing resources appropriately. Future research that explains what is driving cybercrime in these locations might also suggest more appropriate means for tackling the problem. Such an analysis could examine relevant correlates, such as corruption, law enforcement capacity, internet penetration, education levels and so on to inform/test a theoretically-driven model of what drives cybercrime production in some locations, but not others. It also might be possible to make a kind of prediction: to identify those nations that have not yet emerged as cybercrime hubs but may in the future. This would allow an early warning system of sorts for policymakers seeking to prevent cybercrime around the world.

## Limitations

In addition to the points discussed above, the findings of the WCI should be considered in light of some remaining limitations. Firstly, as noted in the methods, our pool of experts was not as large or as globally representative as we had hoped. Achieving a significant response rate is a common issue across all surveys, and is especially difficult in those that employ the snowball technique [59] and also attempt to recruit experts [60]. However, ensuring that our survey data captures the most accurate picture of cybercrime activity is an essential aspect of the project, and the under-representation of experts from Africa and South America is noteworthy. More generally, our sample size (n = 92) is relatively small. Future iterations of the WCI survey should focus on recruiting a larger pool of experts, especially those from under-represented regions. However, this is a small and hard-to-reach population, which likely means the sample size will not grow significantly. While this limits statistical power, it is also a strength of the survey: by ensuring that we only recruit the top cybercrime experts in the world, the weight and validity of our data increases.

Secondly, though we developed our cybercrime types and measures with expert focus groups, the definitions used in the WCI will always be contestable. For instance, a small number of comments left at the end of the survey indicated that the *Cashing out/money laundering* category was unclear to some participants, who were unsure whether they should nominate the country in which these schemes are organised or the countries in which the actual cash out occurs. A small number of participants also commented that they were not sure whether the 'impact' of a country's cybercrime output should be measured in terms of cost, social change, or some other metric. We limited any such uncertainties by running a series of focus groups to check that our categories were accurate to the cybercrime reality and comprehensible to practitioners in this area. We also ran a pilot version of the survey. The beginning of the survey described the WCI's purpose and terms of reference, and participants were able to download a document that described the project's methodology in further detail. Each time a participant was prompted to nominate countries as a significant source of a type of cybercrime, the type was re-defined and examples of offences under that type were provided. However, the examples were not exhaustive and the definitions were brief. This was done partly to avoid significantly lengthening the survey with detailed definitions and clarifications. We also wanted to avoid over-defining the cybercrime types so that any new techniques or attack types that

emerged while the survey ran would be included in the data. Nonetheless, there will always remain some elasticity around participant interpretations of the survey.

Finally, although we restricted the WCI to profit-driven activity, the distinction between cybercrime that is financially-motivated, and cybercrime that is motivated by other interests, is sometimes blurred. Offenders who typically commit profit-driven offences may also engage in state-sponsored activities. Some of the countries with high rankings within the WCI may shelter profit-driven cybercriminals who are protected by corrupt state actors of various kinds, or who have other kinds of relationships with the state. Actors in these countries may operate under the (implicit or explicit) sanctioning of local police or government officials to engage in cybercrime. Thus while the WCI excludes state-based attacks, it may include profit-driven cybercriminals who are protected by states. Investigating the intersection between profit-driven cybercrime and the state is a strong focus in our ongoing and future research. If we continue to see evidence that these activities can overlap (see for example [32, 61–63]), then any models explaining the drivers of cybercrime will need to address this increasingly important aspect of local cybercrime hubs.

## Conclusion

This study makes use of an expert survey to better measure the geography of profit-driven cybercrime and presents the output of this effort: the World Cybercrime Index. This index, organised around five major categories of cybercrime, sheds light on the geographical concentrations of financially-motivated cybercrime offenders. The findings reveal that a select few countries pose the most significant cybercriminal threat. By illustrating that hubs often specialise in particular forms of cybercrime, the WCI also offers valuable insights into the local dimension of cybercrime. This study provides a foundation for devising a theoretically-driven model to explain why some countries produce more cybercrime than others. By contributing to a deeper understanding of cybercrime as a localised phenomenon, the WCI may help lift the veil of anonymity that protects cybercriminals and thereby enhance global efforts to combat this evolving threat.

## Supporting information

**S1 Indices. WCI indices.** Full indices for the WCI Overall and each WCI Type.
(PDF)

**S1 Appendix. Supporting information.** Details of respondent characteristics and analysis of rating behaviour.
(PDF)

## Acknowledgments

The data collection for this project was carried out as part of a partnership between the Department of Sociology, University of Oxford and UNSW Canberra Cyber. The analysis and writing phases received support from CRIMGOV. Fig 1 was generated using information from OpenStreetMap and OpenStreetMap Foundation, which is made available under the Open Database License.

## Author Contributions

**Conceptualization:** Jonathan Lusthaus, Federico Varese.

**Data curation:** Miranda Bruce.

**Formal analysis:** Miranda Bruce, Ridhi Kashyap.

**Funding acquisition:** Nigel Phair, Federico Varese.

**Investigation:** Miranda Bruce, Jonathan Lusthaus.

**Methodology:** Miranda Bruce, Jonathan Lusthaus, Ridhi Kashyap, Nigel Phair, Federico Varese.

**Visualization:** Miranda Bruce.

**Writing – original draft:** Miranda Bruce, Jonathan Lusthaus.

**Writing – review & editing:** Ridhi Kashyap, Nigel Phair, Federico Varese.

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
