## [Decision Letter · Decision Letter 0]

8 Nov 2023

PONE-D-23-32959Mapping the global geography of cybercrime with the World Cybercrime IndexPLOS ONE

Dear Dr. Bruce,

Thank you for submitting your manuscript to PLOS ONE. After careful consideration, we feel that it has merit but does not fully meet PLOS ONE’s publication criteria as it currently stands. Therefore, we invite you to submit a revised version of the manuscript that addresses the points raised during the review process.

We look forward to receiving your revised manuscript.

Kind regards,

Naeem Jan, PhD

Academic Editor

PLOS ONE

Journal Requirements:

Additional Editor Comments (if provided):

Thank you for submitting your manuscript to PLOS ONE. Expert reviewers have carefully reviewed your manuscript and determined that it could be considered for publication in PLOS ONE after a thorough and careful revision. I am therefore inviting you to revise your manuscript according to the reviewers’ comments. Please revise your paper and provide very convincing point-to-point responses according to the comments raised by the reviewers. When you revise your manuscript, please highlight the changes you make in the manuscript by using the track changes mode in MS Word or by using highlighted text.

Reviewers' comments:

Reviewer's Responses to Questions

**Comments to the Author**

1. Is the manuscript technically sound, and do the data support the conclusions?

Reviewer #1: Partly

Reviewer #2: Yes

2. Has the statistical analysis been performed appropriately and rigorously? 

Reviewer #1: Yes

Reviewer #2: Yes

3. Have the authors made all data underlying the findings in their manuscript fully available?

Reviewer #1: Yes

Reviewer #2: Yes

4. Is the manuscript presented in an intelligible fashion and written in standard English?

Reviewer #1: Yes

Reviewer #2: Yes

5. Review Comments to the Author

Reviewer #1: ybercrime is a major challenge facing the world, with estimated costs in the hundreds

of billions. Despite the threat it poses, cybercrime is largely an invisible phenomenon.

Offenders hide behind online nicknames and technical protections, and are dispersed

throughout the world. This means law enforcement faces many obstacles, and existing

technical data is not well suited to establishing the true location of offenders. This

paper proposes a solution: an expert survey with leading cybercrime professionals

from across the world. From March to October 2021 we invited recognized experts in

cybercrime intelligence/investigations to participate in an anonymized online survey on

the geographical location of cybercrime offenders and the severity of their attacks. The

survey asked participants to consider five major categories of cybercrime, nominate

the countries that they consider to be the most significant sources of each of these

cybercrimes, and then rank each nominated country according to the impact,

professionalism, and technical skill of its offenders. The result of the survey is the

World Cybercrime Index, a global metric of cybercriminality organised around five

types of cybercrime. The results indicate that a relatively small number of countries

house the greatest cybercriminal threats. These findings partially remove the veil of

anonymity around cybercriminal offenders, may aid law enforcement and policymakers

in fighting this threat, and contributes to the understanding of cybercrime as a local

phenomenon.

Please see attached file.

Reviewer #2: 1. My suggestion if the paper should address the impact of cybercrime, emphasizing the severity of the issue based on a comprehensive estimation drawn from previous studies.

2. Are there any prior studies similar to this one that employed expert surveys? If so, how does this paper differ from them?

3. Additionally, it's important to outline the limitations associated with using expert surveys.

4. When referring to 'experts,' what criteria have been employed to define them? Are non-technical experts, such as writers, included?

5. Wonder if the survey was conducted in English?

6. During the survey, were there any language barriers encountered?

7. Issues related to biases, as discussed in the Discussion Section, should also be addressed in the Method section.

8. The paper should thoroughly examine the limitations of this study.

6. PLOS authors have the option to publish the peer review history of their article (what does this mean?). If published, this will include your full peer review and any attached files.

Reviewer #1: **Yes: **Timothy C. Haas

Reviewer #2: No

---

## [Author Response · Author response to Decision Letter 0]

22 Dec 2023

Many thanks to the reviewers for their very helpful comments. We have taken these on board and made a number of changes, which have significantly strengthened the paper. Below we highlight each suggestion and how we have addressed it in the text. Line numbers mentioned below correspond to the clean Manuscript file.

Reviewer 1

1) How long is the survey? As you know, there is a literature on respondent fatigue when answering a long survey. Please reference this literature and explain in more detail why you don’t believe that respondents resorted to random answers due to fatigue. See for example, https://www.nber.org/papers/w30439

Controlling the length of the survey was a significant concern during the survey design phase, and we thank the reviewer for prompting us to discuss this in more detail. The duration of the survey for each respondent depended on how many cybercrime categories they addressed, and how many countries they nominated in each of these categories. The maximum number of countries that could be nominated and ranked by a single respondent was 25 countries, which would result in the participant completing a maximum of 75 Likert-types scales. We recognised this as a potential cognitive burden on participants, so we designed the survey accordingly. Based on practice estimates the survey took around 30 minutes to complete, which is not overly long. When we contacted them, we also made it clear to participants to allocate 30 minutes so they could complete the survey in a thoughtful way (and a specific warning was delivered at the beginning of the survey not to undertake it, if they could not dedicate 30 minutes). We now address the issue of survey fatigue in more detail, and assess whether it may have had a significant effect on our results, at lines 216-236. In short, we have evidence that leads us to believe that respondent fatigue did not have a significant effect on the results.

2) How accurate are your cybercrime professionals at correctly identifying the physical location of cybercriminals? I think you need to provide some measure of this accuracy rather than simply relying on the reputation of your respondents because, as you state, correctly identifying the physical location of a cyberattack’s author is very difficult given the nearly untraceable nature of internet traffic.

Ensuring the accuracy of our expert participants’ judgements was an essential aspect of our survey design. We therefore included a number of controls to ensure we received the highest quality responses possible. This included deterring respondents who did not have the time/commitment to offer thoughtful responses, our strict eligibility requirements for experts who have extensive experience deciphering true offender locations, and frequent visual stimulants to ensure participants consider each survey question with care. These controls are included at lines 134-143 and 185-194. In summary, our sampling strategy and survey design is intended to ensure the highest quality responses from our expert pool as possible.

Regardless of the care we took, the reviewer makes an important point that we should be more explicit in verifying the accuracy of the responses we did receive. As stated in the Introduction, previous attempts to accurately map cybercrime hubs have relied on technical data that doesn’t capture the true physical location of offenders. As such, these existing data sources do not make meaningful points of comparison. Instead, we follow Steenbergen and Marks’ (2006) argument that expert judgements should be compared to external measures of the same phenomena to assess their broader validity. While more limited in coverage than the WCI, qualitative cybercrime studies have identified the same key hubs that have attained the higher ranks within the index. This provides strong support for the accuracy of the WCI for the higher ranked countries, and the robustness of the index more broadly. This comparison can be found at lines 422-434. 

3) The manuscript implicitly assumes all cybercrime is conducted by non-state actors: either individuals or by members of criminal networks (organized crime). But it is well-known that many cybercriminals are employed and supported within state-sponsored facilities. Examples include Russia and North Korea. Please explain how your WCI could be used to reduce the amount of cybercrime generated by this population.

We agree with the reviewer that there is an important distinction between non-state and state cyber attacks. This study focuses on profit-driven cybercrime, rather than cybercrime that is motivated by state (or other) interests. Studying state attacks is an important academic concern, but is outside the scope of the WCI. To avoid any confusion, we have made this clearer at line 63 and at lines 190-194.

The broader issue raised by the reviewer, around profit-driven cybercriminals that may be connected to the state in some way, is an interesting and important one. It is correct that some of the countries with high rankings within this index may house profit-driven cybercriminals that are protected by corrupt state actors of various kinds or have other kinds of relationships with the state. We have now made clear that, while the WCI excludes state attacks, it does indeed include profit-driven cybercriminals that might be protected by states. While the WCI has tremendous impact/policy potential, this paper is focused on the empirical presentation and discussion of the index, and is not a policy paper. We agree this policy application is very important, and is the subject of ongoing research, which makes use of the WCI to more directly answer these kinds of questions (see our response to point 4 below). Discussion of state corruption/protection and cybercrime will be a central theme within this future analysis. We have now added discussion in the paper on this issue at lines 526-537.  

4) Many of the discussed approaches to fighting cybercrime implicitly assume these criminals reside in developing countries. But the four top offenders, Russia, China, Ukraine, and the United States are developed countries. What strategies that use your WCI might be effective at turning cybercriminals in these countries away from cybercrime? Specifically, many of these individuals are educated and fully employed.

The reviewer has correctly assessed that one of the WCI’s major contributions is to enable policy-makers to develop well-informed policies targeted at preventing cybercrime. But at this stage, we cannot make any specific recommendations regarding the prevention of cybercrime in specific countries. Answering this question adequately requires formulating a theoretically-driven model of cybercrime hub formation. This model could then be tested with this survey data, along with existing datasets on a series of other indicators (e.g. corruption, law enforcement capacity and so on). This is a very important undertaking, and we are keen to carry out this analysis in the future (now noted in lines 467-470). But this is beyond the scope of the current paper. Similarly, the reviewer’s broader point about the role of economic development in cybercrime hub development is an important one, and will be central to these future theoretical models. There are many different factors that can be used to define a country’s economic status, and determining which of these factors are most relevant to cybercrime hub development will require more analysis. However, we now address the socio-economic diversity of the top cybercrime hubs at lines 460-463 and note what this implies for future research directions and prevention policies.

5) There are no equation numbers.

Equation numbers have now been inserted at lines 252, 259, and 267.

 

Reviewer 2

1) My suggestion if the paper should address the impact of cybercrime, emphasizing the severity of the issue based on a comprehensive estimation drawn from previous studies.

This is a good suggestion. We have now included the more common estimations of cybercrime impact at lines 49-51.

2) Are there any prior studies similar to this one that employed expert surveys? If so, how does this paper differ from them?

Although there are previous small-N qualitative studies that touch on cybercrime geography, using interview data with cybercrime experts, there are no previous studies that have attempted to map the geographical distribution of cybercrime offenders using an expert survey. We have noted this more clearly at lines 111-112.

3) Additionally, it's important to outline the limitations associated with using expert surveys.

We have added further details regarding the limitations associated with expert surveys at lines 485-508. This is an important part of our methodology, and we thank the reviewer for encouraging us to delve into this more deeply.

4) When referring to 'experts,' what criteria have been employed to define them? Are non-technical experts, such as writers, included?

As described in lines 136-138, we define cybercrime experts as “professionals who have been engaged in cybercrime intelligence, investigation, and/or attribution for a minimum of five years and had a reputation for excellence amongst their peers”. 

Only currently- or recently-practicing intelligence officers and investigators were included in the participant pool. We explicitly exclude professionals working in the field of cybercrime research who are not actively involved in tracking offenders, which includes writers and academics. In short, we only include experts with first-hand knowledge of cybercriminals. This was a strict condition in our sampling strategy. We have clarified this point at lines 138-142.

5) Wonder if the survey was conducted in English?

The survey was conducted in English, as were all communications with participants. The manuscript has been edited to clarify this at lines 159-165, and we have included the reasoning for this choice.

6) During the survey, were there any language barriers encountered?

Participants did not report any language barriers or issues in the comment section at the end of the survey, nor in any personal communication. All participants were contacted by email first, which introduced a basic requirement for English proficiency at the earliest stage. This has been clarified at lines 159-160 to include these details. As noted in the text, English is widely spoken by cybercrime experts from across the globe.

7) Issues related to biases, as discussed in the Discussion Section, should also be addressed in the Method section.

We have now included an outline of biases in the Methods section at lines 237-242, in addition to a more thorough discussion of biases throughout the Discussion section.

8) The paper should thoroughly examine the limitations of this study.

The limitations of the study are now discussed in much greater detail in a new Limitations subsection at lines 475-537. We thank the reviewer for prompting us to address the limitations of the study much more explicitly; this has strengthened and clarified the paper’s contribution.

---

## [Decision Letter · Decision Letter 1]

3 Jan 2024

Mapping the global geography of cybercrime with the World Cybercrime Index

PONE-D-23-32959R1

Dear Dr. Bruce,

We’re pleased to inform you that your manuscript has been judged scientifically suitable for publication and will be formally accepted for publication once it meets all outstanding technical requirements.

Kind regards,

Academic Editor

PLOS ONE

Additional Editor Comments

I am happy to inform you that According to the reviewers comments your paper now been accepted for publication in PLOS ONE.

thank You

---

## [Editor Report · Acceptance letter]

19 Mar 2024

PONE-D-23-32959R1 

PLOS ONE

Dear Dr. Bruce, 

I'm pleased to inform you that your manuscript has been deemed suitable for publication in PLOS ONE. Congratulations! Your manuscript is now being handed over to our production team.

Kind regards, 

on behalf of

Dr. Naeem Jan 

Academic Editor

PLOS ONE